# Improving Accuracy and providing Uncertainty Estimations: Ensemble Methodologies for Ocean Forecasting

Ibrahim Hoteit[1], Eric Chassignet[2], Mike Bell[3]

[1]Physical Science and Engineering Division, King Abdullah University of Science and Technology (KAUST), Thuwal, Saudi Arabia
[2]Center for Ocean-Atmospheric Prediction Studies, Florida State University, United States
[3]MetOffice, Exeter, UK

*Correspondence to*: Ibrahim Hoteit (ibrahim.hoteit@kaust.edu.sa)

**Abstract.** Ensemble forecasting has emerged as an essential approach for addressing the uncertainties inherent in ocean prediction, offering a probabilistic framework that enhances accuracy of both short-term and long-range forecasts. By more effectively addressing the intrinsic chaotic nature of mesoscale and sub-mesoscale variability, ensemble methods offer critical insights into forecast errors and improve the reliability of predictions. This paper reviews the ensemble methodologies currently used in ocean forecasting, including techniques borrowed from weather prediction like virtual ensembles and Monte Carlo methods. It also explores the latest advancements in ensemble data assimilation, which have been successfully integrated into both ocean general circulation models and operational forecasting systems. These advancements enable more accurate representation of forecast uncertainties (error-of-the-day) by sampling perturbations conditioned on available observations. Despite the progress made, challenges remain in fully realizing the potential of ensemble forecasting, particularly in developing tools for analyzing results and incorporating them into decision-making processes. This paper highlights the crucial role of ensemble forecasting in improving ocean predictions and advocates for its wider adoption in operational systems.

## 1. Introduction to Ensemble Forecasting

Forecasts of the ocean state generated by numerical models are inherently uncertain owing to the nonlinear chaotic nature and imperfect internal physics of the ocean models, and inevitable uncertainties in their inputs such as initial and boundary conditions, atmospheric forcing, bathymetry, etc. (e.g., Lorenz, 1996; Pinardi et al., 2008; Sandery et al., 2014; Vandenbulcke and Barth, 2015; Kwon et al., 2016; Sanikommu et al., 2020). Thus, the future ocean cannot be completely described by a single forecast model run, and is better described by a set, or ensemble, of forecasts that provides an indication of the range of possible future ocean states and that represents the uncertainty in the forecasts, also known as errors-of-the-day (Houtekamer and Zhang, 2016; Hoteit et al., 2018) (Figure 1).

Ensemble forecasting has increasingly become a key aspect of weather and climate predictions – see Du et al. (2019) for a review – as it provides a basis to communicate forecasts confidence to end users for better decision making. Similarly, it should

become an integral part of ocean forecasts. Ensemble forecasting was indeed proven to provide extended ocean prediction skills compared to deterministic forecasts, especially for extended time-scale predictions (Mullen and Buizza 2002; Ryan et al., 2015). This ensemble probabilistic framework is also needed for short-range forecasting to better describe the intrinsic

chaotic nature of the mesoscale and sub-mesoscale variability resolved by the new generation high-resolution ocean models (Thoppil et al., 2021). Information about forecast uncertainty can be used in many ways. For instance, the probabilistic information that ensembles provide are particularly valuable for early warnings of hazardous conditions in the ocean and can be integrated into the decision-making process based on economic values (Richardson, 2000; Du and Deng, 2010). On short timescales, the probabilistic information is useful to trigger the deployment of environment protection measures in the event

of an oil spill (Barker et al., 2020), to advise fishermen about the most probable regions of fishing zones, to help coast-guards on the probable areas to focus for search and rescue operations (Melsom et al., 2012), or to advise on path planning for autonomous marine vehicles (e.g., Albarakati et al., 2021), etc. On climate time scales, ensemble forecasting is useful for providing probabilistic information on climate indices such as El Nino and the Indian Ocean Dipole (Schiller et al., 2020).

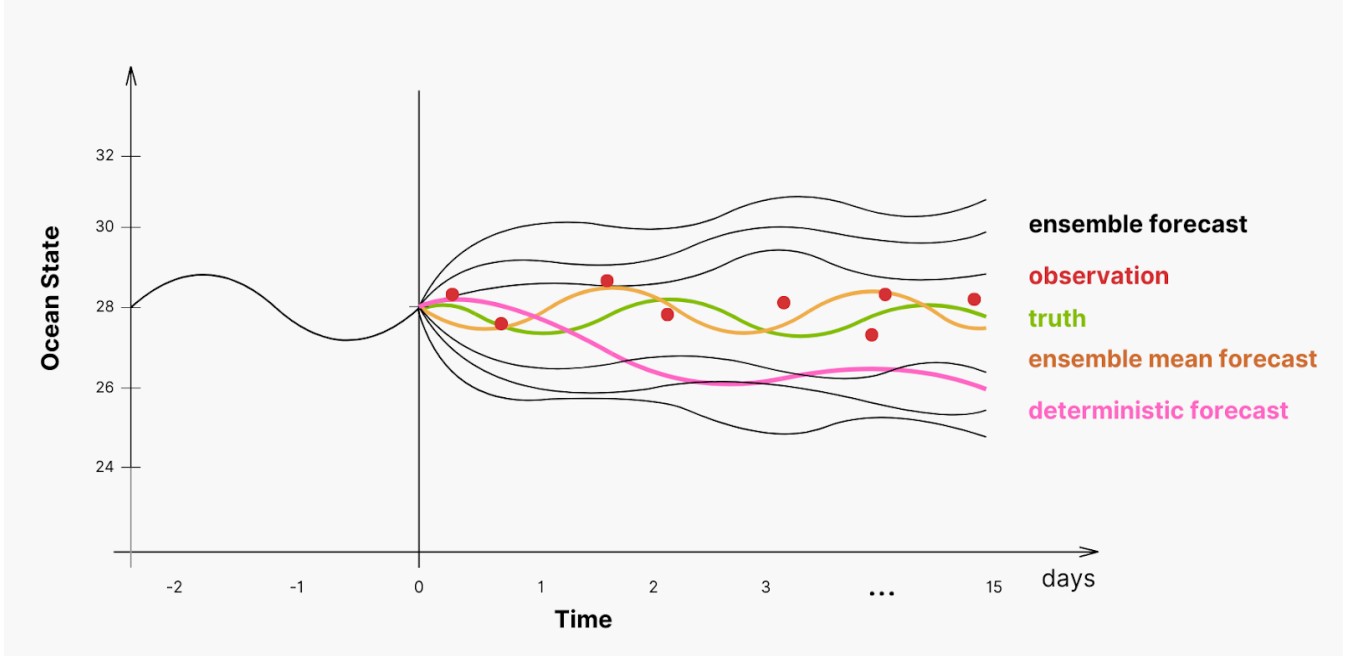

**Figure 1: Schematic illustration of deterministic hindcast (black line at the forecast date 0) and forecast (pink line after day 0), and ensemble forecasts (black lines after day 0) of the ocean state. The ensemble forecasts were driven by various sources of uncertainties (including initial conditions, atmospheric forcing, model physics, bathymetry, …). The ensemble forecast mean and the unknown truth are respectively represented by the orange and green lines. Solid red dots denote observations.**

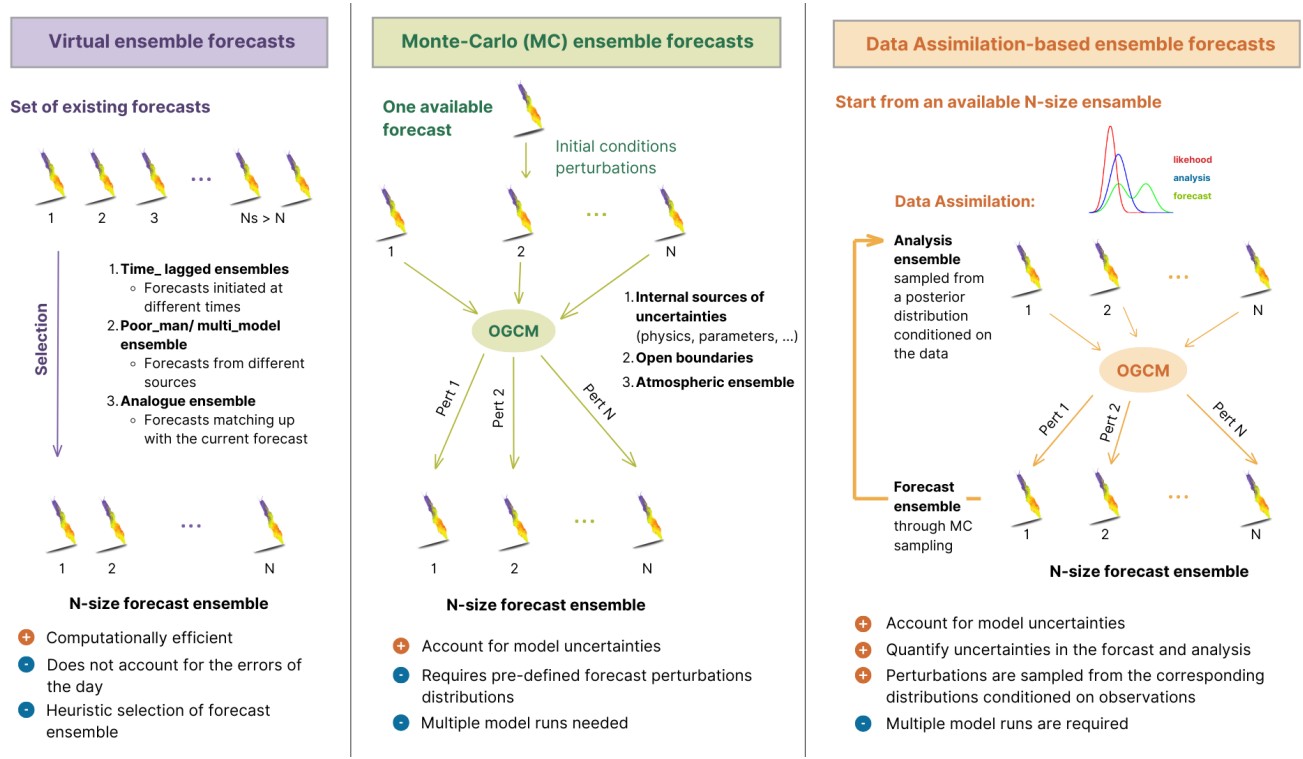

**Figure 2: Schematic diagram illustrating the steps involved in various ensemble forecasting methods. Characteristics of each method are also listed.**

## 2. Methods

Ensemble forecasts find their roots in weather forecasting and can be generated (i) as virtual ensembles whose members are selected from deterministic forecasts and/or historical runs, (Hoffman and Kalnay, 1983; Ebert, 2001; Du, 2004; Schwartz and Sobash, 2017), (ii) or by applying some form of Monte Carlo (MC) analysis in which a set of forecasts are produced by perturbing the model physics and/or inputs, as a way to account for their inherent uncertainties (Martin et al., 2015; Houtekamer and Zhang, 2016; Hoteit et al., 2018). Ensemble forecasts may also be generated following a multi-model approach as the forecasts of different ocean models, or from their combination with MC forecasts (Figure 2). Ideally, the actual future oceanic state should fall within the predicted ensemble range.

- *Virtual Ensemble Forecasts.* The lower-cost virtual ensembles can be used to quantitatively estimate forecast uncertainties based on existing forecasts through various techniques including for instance: (a) the time-lagged ensemble, which automatically creates a forecast ensemble by pulling multiple forecasts that have been initiated at different times, (b) the poor-man ensemble, which gathers single-model forecasts from different sources and is thus a multi-model ensemble from existing forecasts, and (c) the analogue ensemble, made of past forecasts matching up with the current forecast. These methods are straightforward but may result in restricted ensembles due to the limited

available sources of existing forecasts. They are also not designed to capture the flow-dependent error-of-the-day (Du et al., 2019).

- *Monte Carlo (MC) Ensemble Forecasts*. This kind can be generated by perturbing the ocean model physics and/or inputs (Du et al., 2019). Uncertainties in the ocean model could be accounted for by perturbing its internal sources of uncertainties which could come from the missing physics, parameterization schemes, and numerical errors. Different approaches were suggested such as (a) the multi-physics approach which uses a different parameterization scheme for each ensemble member (Sanikommu et al., 2020), (b) the perturbed parameters approach of a selected parameterization scheme, and (c) the stochastic parameterizations approach which injects stochastic perturbations into the physical parameterization schemes (Brankart et al., 2015; Storto and Andriopoulos, 2021). Additionally, given that the short-term predictability of the atmosphere and the ocean is dominated by their initial conditions (ICs), various methods to perturb the initial model state have been proposed to generate ensembles. These include (i) random perturbations sampled from some available error statistics, (ii) the singular vectors and their variants designed to represent the perturbations with the fastest error growth, and (iii) the vector breeding approach, which computes the initial perturbations as the differences between a pair of past concurrent forecasts. Different approaches were also suggested to perturb the bathymetry, open boundaries and the atmospheric and river forcing (Lima et al., 2019; Storto and Yang 2023; Zedler et al., 2023), but ensembles of atmospheric and oceanic forecasts are now available from the global operational prediction centers and can be readily used to generate ocean forecasts ensembles.

- *Data Assimilation-Based Ensemble Forecasts*. Ensemble forecasts in data assimilation are typically generated by introducing multiple, slightly different estimates of the current system state to capture uncertainties in observations and model parameters while accounting for the "error-of-the-day". For example, in an Ensemble Kalman Filter (EnKF), observations can be perturbed (or not) (Whitaker and Hamill, 2002; Hoteit et al., 2015), and the model is then integrated from these perturbed initial states, sampled according to the estimated initial-state statistics derived from previous forecasts and the most recent observations, resulting in an ensemble of forecasts. Additional perturbations may be introduced to the model physics or inputs to represent other sources of uncertainty, as demonstrated in Monte Carlo (MC) ensemble forecast methods (Whitaker and Hamill, 2012; Hoteit et al., 2018; Sanikommu et al., 2020). This collection of forecasts provides a probabilistic picture of future conditions, reflecting both initial condition and model uncertainties.

Virtual ensemble forecasts were traditionally more common for operational purposes as they do not require major extra computations, although their large ensemble spread was perceived as a disadvantage. The multi-model approach involves the tedious task of running and maintaining different ocean general circulation models (OGCMs), but it can be facilitated by combining the forecasts from different operational centers (e.g., Ren et al., 2019). Ensemble forecasts generated by a MC approach are increasingly adopted operationally. Despite their demonstrated skill, the MC ensemble forecasts require that the ensemble truly represent the probability distribution of the underlying dynamical system (Leith, 1974). Designing perturbation

schemes that accurately capture all sources of uncertainty (e.g., initial conditions, forcing, model physics) remains a significant challenge, as does determining how to vary these perturbations in time.

Recent advances in ensemble data assimilation approaches now provide robust frameworks to represent the error-of-the-day, for both initial conditions and inputs or parameters, by sampling perturbations directly from (approximate) error distributions conditioned on observations (Hoteit et al., 2018; Carrassi et al., 2022). Nevertheless, obstacles persist, particularly in high-dimensional ocean forecasting systems, where the ensemble size is often limited by computational costs. Methods such as localization and inflation are commonly used to mitigate sampling errors and maintain adequate ensemble spread (Brankart et al., 2015; Storto and Andriopoulos, 2021). Hybrid ensemble–variational approaches and other advanced techniques can further alleviate these issues by blending flow-dependent ensemble covariances with multi-year or climatological statistics (Buehner, 2010). However, each solution carries its own computational demands and assumptions, highlighting the ongoing need to balance accuracy, efficiency, and complexity in operational ocean forecasting systems.

## 3. Probabilistic assessment

Forecast ensembles are evaluated through their sample statistics, mainly the ensemble mean and its spread (the standard deviation with respect to the ensemble mean). The mean can be directly compared with available observations, while the spread indicates the confidence in the forecast: a smaller spread implies lower uncertainty, and vice versa. High-order moments, such as skewness and Kurtosis, help characterize the shape of the ensemble distribution (Groeneveld and Meeden, 1994). In addition, probabilistic validation and verification methods, including reliability, resolution, sharpness and rank histograms, are frequently employed (Johnson and Bowler, 2009). An ensemble is deemed reliable if the predicted probability of an event aligns with the observed frequency. Resolution assesses how far the forecast deviates from the climatological event frequency; increasing this deviation enhances the reliability of the forecast. In the same context, sharpness measures the ability of an ensemble forecast to spread away from the climatological average. Ideally, an ensemble forecast needs to be reliable, with as many forecasts as possible away from the climatological average. Rank histograms, which tally the position of the observation among sorted ensemble values, are used to test reliability and diagnose errors in the ensemble mean or spread (Hamill, 2001). Another commonly used metric is the Continuous Ranked Probability Score (CRPS), which evaluates both accuracy and reliability by comparing the forecast distribution with the observed value across all possible outcomes. A lower CRPS indicates a closer match to reality and thus better overall probabilistic forecasts (Leutbecher and Haiden, 2021).

## 4. Current status of ensemble forecasts in Operational Ocean Forecasting Systems (OOFSs)

Despite the early establishment of ensemble methods for ocean data assimilation and forecasting (Evensen, 1994), ensemble forecasts, particularly the global systems, only recently found their ways to the operational centers. This is mostly because the centers prioritized using the available computational resources to increase the resolution of ocean models. This was due to the need to resolve the mesoscale to sub-mesoscales processes to better describe the energy cascade in the ocean, and to meet user demands for higher resolution forecasts (e.g. D'addezio et al., 2019; Davidson, 2021). Recent developments in ocean ensemble

forecasting followed the improved coverage in ocean observations that provided increased information to accurately constrain the initial ocean state for extended forecast horizons, the better coordination between ocean forecasting groups, the ease of access to atmospheric ensembles, and the ever-increasing availability of computational power (Metzger et al., 2010; Smith et al., 2011; Strohmaier et al., 2015; Bauer et al., 2021). Ocean ensemble forecasts are now routinely generated at several operational ocean centers on both global and regional scales to cater to different needs as summarized in Table 1.

**Table 1: Summary of selected operational ensemble forecasting systems worldwide.**

| Institution | Forecasting System | Domain (resolution) | Ensemble Perturbations (Size) | Type of Forecast | Reference |
|---|---|---|---|---|---|
| Met Office, UK | FOAM | Global (9km) | Observations + internal physics + Atmosphere (36) | Short-range ocean state | **Lea et al. (2022)** |
| NRL, USA | Navy-ESPC | Global (9km) | Observations (16) | Days to subseasonal ocean state | **Barton et al. (2021)** |
| Bluelink, Australia | OceanMAPS | Global (10km) | Initial conditions + Time-lagged (48) | Short-range ocean state | **Brassington al. (2023)** |
| ECMWF | NEMO | Global (25km) | Initial conditions + Forcing + Observations (5) | Near Real time ocean state | **Zuo et al. (2019)** |
| NERSC, Norway | TOPAZ5 | North Atlantic and Arctic (6km) | Atmosphere (100) | Short-range ocean state | **Nakanowatari et al. (2022)** |
| KAUST, Saudi Arabia | MITgcm | Red Sea (4km) | Atmosphere + Internal physics (50) | Short-range ocean state | **Sanikommu et al. (2020)** |
| INCOIS, India | ROMS | Indian Ocean (8km) | Atmosphere + Internal physics (80) | Short-range ocean state | **Balaji et al. (2019)** |

| | | | | | |
|---|---|---|---|---|---|
| Bureau of Meteorology, Australia | ACCESS-S | Global (4km) | Internal physics + Time-lagged (30) | Multi-week to seasonal ElNino/IOD | **Wedd et al. (2022)** |
| CMA, China | CMMEv1 | Global (100km) | Multi-model + Initial conditions (90) | Multi-week to seasonal ElNino/IOD | **Ren et al. (2019)** |
| CMCC | CMCC-SPS3.5 | Global (25km) | Initial conditions + Model physics (50) | 184 days | **Gualdi et al. (2020)** |
| ECMWF | SEAS5 | Global (25km) | Initial conditions + Model physics + Observations (51) | 6 months | **Johnson et al. (2019)** |
| Meteo-France | Meteo-France System 8 | Global (25km) | Model dynamics (51) | 7 months | **Pianezze et al. (2022)** |
| DWD | GCFS 2.1 | Global (25km) | Initial conditions + Model physics (50) | 215 days | **Frohlich et al. (2021)** |
| ECMWF | IFS | Global (10km) | Internal physics (51) | Short-range waves | **Browne et al. (2019)** |
| NCEP | GWES | Global (25km) | Wind (30) | Short-range waves | **Penny et al. (2015)** |
| UK Metoffice | Wavewatch-III | Atlantic-UK (3km) | Wind (22) | Short-range waves | **Bunny and Saulter (2015)** |
| MET-Norway | Barotropic version of ROMS | Norway (4km) | Atmosphere (51) | Short-range storm surge | **Kristensen et al. (2022)** |

## 5. Role of ensemble forecasts in next generation OOFSs

Recognizing the importance of representing uncertainties in ocean forecasts to meet the need of future demands in probabilistic predictions, ensemble forecasts are expected to become a standard output of any operational ocean product. Although high-resolution observations of some surface variables are now more accessible, the lack of dense, three-dimensional coverage, especially at subsurface levels, still leaves mesoscale and submesoscale processes poorly constrained by ocean analysis systems. Uncertainties from the unconstrained scales might lead to larger forecast errors due to growing dynamical instabilities (Sandery et al., 2017), which limits the forecasting skills of high-resolution ocean models (e.g., Thoppil et al., 2021). Ensemble forecasting has been proven efficient to extend ocean forecasting horizons when model uncertainties in the initial conditions, inputs, and physics are accounted for (Mullen and Buizza 2002; Ryan et al., 2015; Sanikommu et al., 2020). Ensemble forecasts are also essential for providing the error statistics required by ocean analysis systems, thereby enabling better use of high-density observations from recently launched and upcoming satellite missions, such as Surface Water and Ocean Topography (SWOT) (Fu and Ubelmann, 2014). Long delayed by the desire of the community to increase the resolution of the ocean models to improve their realism, the ever-increasing computing resources will provide more and more power to integrate these within ensemble forecasting frameworks.

Ocean forecasts have long been produced by data assimilation (DA) systems and are now routinely used operationally. Ensemble forecasts could be generated from deterministic DA systems, which produce one single-forecast, by simply perturbing the observations (or other parameters of the assimilation system), or during the forecasting step using an ensemble forecasting method. Ensemble DA methods, on the other hand, readily produce ensemble ocean perturbations that (approximately) represent the error-of-the-day and can be directly used to generate ensemble forecasts. These could be also combined with standard ensemble forecasting methods to further represent the missing information about the error growth in the computationally restricted DA ensembles. To fully exploit the benefits from ocean ensemble forecasts, new tools to analyze and visualize, and also integrate these probabilistic products in decision making and management of ocean services need to be developed and made available for the end users.

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

**Competing interests**

The contact author has declared that none of the authors has any competing interests.

**Data and/or code availability**

This can also be included at a later stage, so no problem to define it for the first submission.

**Authors contribution**

This can also be included at a later stage, so no problem to define it for the first submission.

**Acknowledgements**

The authors greatly acknowledge the contributions of Dr. Siva Reddy and Dr. Naila Raboudi to this article and for their invaluable insights.