# Peer review of "Improving Accuracy and providing Uncertainty Estimations: Ensemble Methodologies for Ocean Forecasting"

_State of the Planet, 2024_

## Author Comment (AC1)

**Replies to Reviewer #1's Comments**

The paper provides a brief description of the motivation for producing ensemble ocean forecasts together with an overview of the different methods for generating them. A brief description of how ensembles are assessed is also provided. A table is included which lists some existing operational ensemble ocean forecasting systems. The paper is a useful introduction to the methods for ocean ensemble forecasting and the status of the field.

*We would like to thank the Referee for the time and effort invested in reviewing our manuscript, as well as for the valuable comments and suggestions provided. All points raised have been addressed in the revised version, and below we offer point-by-point responses to each comment*

- *Main comments*

The structure of the text in section 1.1 could be improved. The text doesn't seem to clearly follow the different options for ensemble production as shown in Fig. 2.

*Thank you for pointing this out. We have incorporated the following discussion on DA-based ensemble forecasts in Section 2 (previously Section 1.1):*

*"Data Assimilation-Based Ensemble Forecasts. Ensemble forecasts in data assimilation are typically generated by introducing multiple, slightly different estimates of the current system state to capture uncertainties in observations and model parameters while accounting for the "error of the day". For example, in an Ensemble Kalman Filter (EnKF), observations can be perturbed, or not, Whitaker and Hamill, 2002; Hoteit et al., 2015), and the model is then integrated from these perturbed initial states, sampled according to the estimated initial-state statistics derived from previous forecasts and the most recent observations, resulting in an ensemble of forecasts. Additional perturbations may be introduced to the model physics or inputs to represent other sources of uncertainty, as demonstrated in Monte Carlo (MC) ensemble forecast methods (Hoteit et al., 2018; Sanikommu et al., 2020). This collection of forecasts provides a probabilistic picture of future conditions, reflecting both initial condition and model uncertainties."*

More references could be included to give the reader more information about particular advances, as suggested in the minor comments below.

*We have carefully considered all of the references suggested by the Reviewer and incorporated them into the revised manuscript at the appropriate points. Thank you.*

Table 1 is quite an ad hoc selection of different systems and not all up to date. Some suggestions for improving it are included in the minor comments below.

*We have cited, to the best of our knowledge, additional systems as suggested by the Reviewer. Thank you.*

- *Minor comments*

The section numbering seems quite ad hoc. Perhaps all sub (and sub-sub) sections could be made into new main sections.

*We have carefully reviewed and updated our section structure to ensure better clarity and coherence.*

Line 31. Suggest rewording to "…to communicate forecast confidence to end users for better decision making."

Done. Please refer to line #25 in the revised manuscript.

Figure 1 caption. Not sure why SST is specified as the observation type – the rest of the Fig and caption are more general.

Thank you for bringing this to our attention. We have removed the mention of "SST" from the figure caption to better reflect its general context, as suggested by the Reviewer.

Line 58. Fig. label is incorrect.

Corrected. Thank you.

Line 74. "Alternatively, …". The initial condition uncertainty is an additional source of uncertainty rather than an alternative to those mentioned previously.

Thank you for pointing this out. We have revised the text to clarify that the initial condition uncertainty is an additional source of uncertainty (please see line #68 in the revised manuscript). We appreciate your helpful suggestion.

Line 70. Some references for ocean model stochastic model schemes would be useful here, e.g. from Storto et al., 2021, Brankart et al., 2015.

Thank you for suggesting these references. We now cite them in the revised manuscript (please see line #68).

Line 76. It wasn't clear to me how EDA schemes fit into these options.

This short article is focusing on ensemble forecast methods, whether the perturbations are heruistic, dynamical, or from EDA. We are hoping that the new paragraph we included on DA should address the Reviewer's comment.

Line 79. Lateral boundary and surface forcing perturbation schemes might still be needed for some applications where the available atmospheric and global ocean ensembles may not be appropriate in a given operational setting. Perhaps some references could be included on these, e.g. Storto et al., 2023. You could also mention about the uncertainty in other inputs such as the rivers, e.g. Zedler et al., 2023.

Thank you for the suggestion. We now mention other sources of uncertainties including bathymetry and rivers forcing. We have also incorporated all the suggested references into the revised manuscript (please refer to line #74-75).

Line 100. You don't mention CRPS here which is often used to assess ensemble forecasts.

We have incorporated information about CRPS (please refer to line #118-120 in the revised manuscript). Thank you for the suggestion.

Line 110. Do you mean global systems here? Some regional operational forecasting systems have been running ensembles for a long time, e.g. TOPAZ (Bertino et al., 2008).

Thank you for drawing our attention to this detail. We have clarified in the revised manuscript that this statement is particularly applicable to global systems (please see line #25).

Line 115. Missing comma between "horizons" and "the".

Corrected. Thank you.

Line 119. Perhaps it could be stated that table 1 is a selection of systems, rather than a comprehensive list (which is difficult to provide).

Thank you for the suggestion. We have corrected the caption as suggested. It reads now: "Table 1: Summary of selected operational ensemble forecasting systems worldwide."

Table 1:

The FOAM ensemble includes internal physics perturbations according to Lea et al. 2022.

The Bluelink system now runs an operational 1/10° global ensemble using the EnKF (Brassington et al., 2023).

The ECMWF ocean system is an ensemble system (Zuo et al., 2019) and could be included.

A couple of surface wave systems are listed, but others also run ensemble wave forecasts, e.g. MeteoFrance, UK Met Office.

Seasonal forecast systems are mostly ensemble-based systems but only the BoM system is listed. See https://climate.copernicus.eu/seasonal-forecasts.

A regional system that could also be included is described by Röhrs et al., 2023.

We appreciate the Reviewer's observation regarding the difficulty of listing all operational systems. Consequently, we have updated the table to include several additional operational systems, as suggested.

Line 130. SWOT is now flying so this sentence should be amended.

Thank you for the update regarding the SWOT mission. In the revised manuscript, we have amended the sentence to acknowledge that SWOT is now operational. It reads:

"Ensemble forecasts are also needed to provide error statistics for the ocean analysis systems to better exploit the high-density observations from upcoming and newly operational satellite missions, such as Surface Water Ocean Topography (SWOT) (Fu and Ubelmann, 2014)."

Line 133. Ocean forecasting systems have been produced by DA system for a long time so I wasn't quite sure why this sentence was included.

We have revised this sentence to "Ocean forecasts have long been produced by data assimilation (DA) systems and are now routinely used operationally".

---

## Author Comment (AC2)

**Replies to Reviewer #2's Comments**

We appreciate the Referee's thorough review and constructive feedback. We have carefully addressed all comments and incorporated the necessary changes in the revised manuscript. Below, we provide point-by-point responses to each of the Reviewer's suggestions.

In this section 1.2 on probabilistic assessment, the explanation of the different diagnostics is not so easy to understand for someone who is not familiar with ensemble approaches. Few additional sentences illustrating the metrics would help to understand their meaning for non experts. I would also suggest to add references for readers interested in more details (for example: Section 12.B Statistical Concepts - Probabilistic Data - Forecast User Guide - ECMWF Confluence Wiki).

We thank the Reviewer for this helpful suggestion. In the revised manuscript, we have included additional clarifications of these metrics and cited further references for easier understanding.

Some additional information on the limitations, challenges for ensemble approaches and the use of hybrid methods in OOFSs would be useful to describe in this article to give a more complete and objective point of view. For example, does the "inflation" methods still need to be used or regularisation / localisation in case to small ensemble, …?

Thank you for highlighting the limitations and challenges associated with ensemble methods. While our article aims to offer a concise overview of ensemble forecast status, space constraints prevent us from addressing every challenge in detail, including issues such as inflation methods, regularization, and localization for small ensembles, which are more pertinent to ensemble data assimilation methods. In response to the reviewer's suggestion, we have added a short discussion on the auxillary techniques to mitigate for the loss in (small) ensembles spread and on hybrid methods (please refer to our reply to Comment 1.94). For those interested, relevant references are included to offer further insight.

The numbering of the different sections of the article needs to be revised: 1, 1.1, 1.2, 1.1.1, 2. The section 1.1.1 may be changed to 1.3.

We have changed all the Subsections to main Sections as suggested.

*Line by line comments*

Figure 2:

Virtual ensemble: for point 2. "Source" may be to vague: replace by model/system as in the text?

 "Source" is replaced by "System".

"Adhoc selection": I do not clearly understand what it means.

We replaced it by "heuristic selection".

DA ensemble forecast: quantity -> quantify ?

Done. Thank you.

l.89: can you give few more details/references on the present limitations and method to overcome them to illustrate today limitations and challenges that still need to be addressed or recognized as a limitation for ensemble forecasts and ensemble DA analysis.

We thank the Reviewer for this suggestion. We have expanded our discussion in the revised manuscript to include additional details on the current limitations of ensemble forecasting and ensemble data assimilation. In particular, we highlight computational constraints, the challenges of designing representative perturbation schemes, and the trade-offs between accuracy and efficiency. We also provide references to methods such as localization, inflation, hybrid ensemble-variational approaches, and other advanced techniques that can help mitigate these challenges.

l.94: Hybrid methods or multi-scale analysis, with the use of lower resolution ensemble covariance than the HR analysis, should at least be mentioned even if not discussed here.

Thank you for the suggestion. We now mention the hybrid methods in the revised manuscript, as recommended. However, we have not expanded on multi-scale analysis, given that our primary focus is on ensemble forecasts rather than the intricacies of ensemble data assimilation. We hope this addresses the Reviewer's comment while allowing the manuscript to remain aligned with its main objectives.

l.109: OOFS acronym is not defined.

Thank you for pointing this out. We have now defined OOFS in line number 125 of the revised manuscript.

l.112: The increased/high resolution "tendency" is also to answer the user requests for higher resolution analysis and forecasts.

We have revised this sentence as suggested: "This was due to the need to resolve the mesoscale to sub-mesoscales processes to better describe the energy cascade in the ocean, and to meet user requests for higher resolution forecasts (e.g. D'addezio et al., 2019; Davidson et al., 2021)."

l.117: ever-increasing availability of computational power? Any more recent references than 2010 and 2011 to support it?

We have provided two more recent references following the Reviewer's comment.

l.119: table 1: As it is the view at a given date you may need to write it explicitly and mention that the table does not show the ensemble analysis and forecasting systems are under development in OOF centers.

The Table caption was revised for more clarity and reads now as "Summary of key operational ensemble forecasting systems worldwide".

Table 1: It would be nice to add the spatial resolution. In some OOF centers, the ensemble "system" is complementarity to a deterministic higher resolution system.

Thank you for the suggestion. We now report the spatial resolution of these models in Table 1.

l.123: Lack of physical HR data: There is today high-resolution satellite observations that are not fully exploited with fine scale observed as some SST, Ocean Colour products and now SWOT observations also going very close to the shore. The lack of HR data is true for the ocean interior.

We thank the Reviewer for this valuable comment. While there are indeed high-resolution observations available (e.g., from satellite SST, ocean color products, and SWOT), the coverage remains limited to only a few variables. In particular, obtaining high-resolution data for sea surface salinity and currents is still challenging, and, as the Reviewer correctly point out, the availability of such data is even more limited in the ocean interior. We have revised the sentence to clarify this limitation:

"Although high-resolution observations of some surface variables are now more accessible, the lack of dense, three-dimensional coverage, especially at subsurface levels, still leaves mesoscale and submesoscale processes poorly constrained by ocean analysis systems."

l.124: "poorly constrained" and not "poorly unconstrained"?

Corrected. Thank you.

l.130: Since the SWOT data are now available, the text must be updated.

We have revised the sentence to reflect the update. It reads now:

"Ensemble forecasts are also essential for providing the error statistics required by ocean analysis systems, thereby enabling better use of high-density observations from recently launched and upcoming satellite missions, such as Surface Water and Ocean Topography (SWOT) (Fu and Ubelmann, 2014)."

---

## Author Comment (AC3)

**Replies to Reviewer #3's Comments**

We are grateful for the Referee's thorough review and constructive feedback. We have carefully addressed all comments and revised the manuscript accordingly. Below, we provide detailed, point-by-point responses to each of the Reviewer's suggestions.

We would like to thank the authors for opening a much-needed discussion on ocean ensemble forecasting. In particular, I find their classification of types of ensemble initialisations very useful. If I may, I would like to add a point here about a type of ensemble that could be coined as identity-retaining ensembles. The idea is the following: While the Monte-Carlo ensemble initialization relies on explicit perturbations, the data assimilation informed ensemble derives ensemble increments for each member and thereby maintains spread. Ensemble forecast cycles are initialized such that each ensemble member is initialized by a forecast of the same member from the previous cycle, removing the need for explicit perturbation. This is the case in EnKF applications, but even without any data assimilation an identity-retaining ensemble is perceivable that describes flow-dependent uncertainty with an error of the day, potentially differentiating predictable regimes from unpredictable circulation patterns.

To provide more context, I would like to suggest that the need for initialization in ocean prediction is discussed in more detail. A comparison with atmospheric prediction that lies the scientific foundation for ensemble forecasting could be useful here. An important difference is that weather prediction benefits from very accurate analysis, and the scope of the EPS is to model the uncertainty due to unstable growing modes which can be adequately described using Monte-Carlo type perturbations. Ocean ensembles, on the other hand, need to deal with large uncertainties in the initial conditions, driving the need for approaches that are different to what is needed in weather prediction, as e.g. identity-retaining ensembles.

Thank you for these insightful suggestions. In the revised manuscript, we have expanded our discussion of the need for initialization in ocean prediction, explicitly discussing ensemble data assimilation methods. We recognize that atmospheric prediction provides important foundational concepts for ensemble forecasting; however, we have chosen to maintain our focus on ocean ensembles to avoid diluting the core scope of this work. We appreciate the Reviewer's feedback and trust the additional details will offer clearer insight into the specific challenges and strategies involved in ocean prediction

With regard to the listed ocean ensembles, I'd like to point to the comments of reviewer #1, adding to the overview list of operational ocean ensemble systems. Perhaps a sorting of this list with respect to forecast variables could be useful, e.g. waves, storm surge, ocean circulation/ hydrodynamics, and sea ice, and a remark on how these systems fit into your classification of ensemble initializations.

We have revised the table following the Reviewer's suggestions (and those of Reviewer #1), by incorporating more operational centers and sorting the list with respect to forecast variables. Thank you.

Please note that the Topaz 4 model is not operational any more and has been replaced by Topaz5; https://data.marine.copernicus.eu/product/ARCTIC_ANALYSISFORECAST_PHY_002_001/de scription.

Corrected. Thank you.